# Designing Composite Adaptive Propeller Blades with Passive Bend–Twist Deformation for Periodic-Load Variations Using Multiple Design Concepts

**DOI:** 10.3390/polym15122749

**Published:** 2023-06-20

**Authors:** Sondre Østli Rokvam, Nils Petter Vedvik, Luca Savio, Andreas Echtermeyer

**Affiliations:** 1Department of Mechanical and Industrial Engineering (MTP), Norwegian University of Science and Technology (NTNU), 7491 Trondheim, Norwayandreas.echtermeyer@ntnu.no (A.E.); 2SINTEF Ocean, Postboks 4762 Torgarden, 7465 Trondheim, Norway; luca.savio@sintef.no; 3Kongsberg University Technology Centre “FleksProp”, NTNU, 7491 Trondheim, Norway

**Keywords:** composites, bend–twist, propellers, design, FEA

## Abstract

Four plausible design concepts are applied together to investigate composite bend–twist propeller-blade designs that show high twisting per bending deflection. The design concepts are first explained on a simplified blade structure with limited unique geometric features to determine generalized principles for applying the considered design concepts. Then, the design concepts are applied to another propeller-blade geometry to obtain a bend–twist propeller-blade design that achieves a specific pitch change under an operational loading condition with a significant periodic-load variation. The final composite propeller design shows several times more bend–twist efficiency than other published bend–twist designs and shows a desirable pitch change during the periodic-load variation when loaded with a one-way fluid–structure-interaction-derived load case. The high pitch change suggests that the design would mitigate undesirable blade effects caused by load variations on the propeller during operation.

## 1. Introduction

### 1.1. Ship Propellers

Many large ships use marine propellers for propulsion. Propeller designs primarily must show good hydrodynamic properties, such as propulsive thrust and efficiency. However, other properties, such as margin of cavitation, noise signature, manufacturability, strength and longevity, must also be considered [1]. The wide range of properties desired in a propeller design introduces possible conflicts between opposing requirements from different disciplines, making it necessary to compromise between constraints and aims to achieve well-designed propellers [2].

Traditionally, rigid solid cast machined isotropic metal propellers have been the norm for marine propellers. For such rigid propellers, the propeller’s geometric shape is the main factor when designing propellers for hydrodynamic performance [1]. However, a more recent alternative approach to propeller designs has been to engineer adaptive polymer-based composite blades with a bend–twist coupling that shows favourable blade deformation, which can improve fluid dynamic performance [1,3,4,5,6,7,8,9,10,11,12,13,14,15,16,17,18,19,20]. Bend–twist coupling has also been applied to composite products such as wind turbines and tidal turbines [4,6,16,21], and much of that experience can be applied to propeller blades. Furthermore, polymer-based composite materials have been said to have the potential to rival metals in propeller application; however, neither material can be said to be superior in light of today’s technology [18].

The concept of tailoring polymer-composite material through a reinforcement ply orientation to get anisotropic layups for adaptive fluid elastic propellers has been around since the 1970s in composite prototyping [22,23]. For marine composite propellers, the field was rejuvenated by one-way and later two-way fluid–structure interaction (FSI) modelling in the 1990s–2000s [19,20,24]. (An explanation of the difference and application of one-way and two-way FSIs is provided in Section 2.3.1). At approximately the same time, the optimisation of ply orientation in anisotropic layups for composite propellers was shown [25]. While an extraordinary feat, such algorithms are only optimised within the considered parameters and for the considered desirable outcome. However, how one can arrange various materials to make a composite propeller is mainly limited by the materials, blade geometry, imagination and manufacturability [8]. The vast number of factors that need to be considered makes the adaptive propeller-blade design process complex. Designing performance-enhancing elastic response and deformation characteristics in composite products has been conducted in wind-turbine blades, helicopter-rotor blades, tidal blades, and marine propeller blades, proving the concept [3,5,6,9,11,12,13,14,16,26,27,28,29]. Publications experimentally verifying fluid–structure interaction analyses have helped justify the feasibility of state-of-the-art numerical simulations in flexible propeller designs [30,31,32].

In practical use, adaptive blades can, for example, be useful in operating scenarios where the propeller blades are unequally loaded. Unequal loading on the different blades causes periodic loading on each propeller blade as the propeller revolves for propulsion. This periodic loading on the blades causes an unbalanced rotation in the propeller system with undesired effects, such as vibrations and noise, which affect the durability of the propeller system and can be harmful to marine life [33]. For propellers mounted on marine thrusters for manoeuvrability improvement, unbalanced rotation is a common problem [10,34,35]. When the thruster and propeller operate at a steering angle to change the ship’s heading, the total loads on the propeller blades and the thrust contribution from each blade can vary by a significant factor periodically throughout every propeller revolution. The large load variation means that one blade can be overloaded and close to failure, while another blade is hardly loaded or contributing to the propulsion at the same time.

### 1.2. Bend–Twist Coupling

Adaptive composite propeller blades with a bend–twist coupling that reduces the pitch at higher loads have been shown to mitigate unbalanced rotation in propellers. In bend–twist coupling, twist deformation is a secondary effect of bending deflection. The twisting of a deflecting blade causes a pitch change in the blade that functions similarly to controllable pitch propellers but is an instant material response with no active control needed. The desired way to twist the blade for a load shed-off at higher loads is to decrease the blade’s pitch to reduce the lift coefficient and the hydrodynamic foil lift in the blade. Thus, as the blade load increases, the blade deflects from the load but twists the blade into a lower pitch, relatively lessening the blade load hydrodynamically. Researchers have already successfully designed, built, and verified composite propeller blades with such properties [5,12,13]. With such deformation characteristics, the load variation on the propeller blades is mitigated as the blades loaded with an above-average load in a periodic-load case twist to decrease their pitch and their load, while the blades loaded at below the average load have a relatively more aggressive pitch.

A recent review paper by Young et al. envisioned that high-tech sophisticated commercial adaptive composite ship propeller blades might be developed in the foreseeable future, possibly with sensors for in-situ structural health monitoring and active electromechanical controls based on fast-moving piezoelectric actuators and compliant structures to actively control the blade shape [3]. Nevertheless, there are many requirements that commercial adaptive composite propeller blade designs need to balance, including fundamental features such as the hydrodynamic shape of an outer surface, a solution for the connection to the hub, and sufficient strength and durability, which must also undergo iterative prototype development with the available manufacturing technology when designing and manufacturing composite propeller blades. However, as the concept has been proven, the other intriguing aspect of adaptive bend–twist propellers is the engineering challenge of how well they can be built.

The structural composition of traditional solid isotropic metal propellers is straightforward—solid metal. In structural buildups of adaptive composite marine propellers in the published literature, the focus has primarily been on hollow-structure and sandwich-structure designs of composite materials for bend–twist coupling in anisotropic laminates. Simultaneously, in other flexing foil products, such as wind turbine blades or plane wings, it is common to use a substructure system of stiffeners, ribs and spars of various materials, including polymers and reinforced polymers, to reinforce and guide or limit deformation.

Marine propellers are typically relatively short compared with the long and slender blades of tidal turbines, helicopter rotors, and wind turbines. Due to the short span, obtaining an efficient bend–twist coefficient is vital to obtaining enough twist deformation in the short blades. Therefore, applying multiple design concepts to obtain bend–twist coupling in marine propeller blades might be a helpful toolbox expansion towards high bend–twist composite propeller-blade designs. In addition, composite material products often consist of many parts and components, some of which could be metal. Thus, hybrid designs with subcomponents in various materials are also possible and considered in this paper.

### 1.3. Approach and Novelty

This paper expands on the explicit adaptive propeller-design concepts in prior publications by presenting a new design process that applies multiple plausible design concepts, both common and novel, for bend–twist coupling in a propeller-blade design process. The final design is engineered to mitigate periodic-load variations caused by turned-thruster operational scenarios. In addition, a novel quantitative analysis of the deformation characteristics for different design concepts is performed to explicitly estimate the potential bend–twist contributions achievable within the considered design concepts.

The final bend–twist propeller-blade design in this paper is applied to a typical propeller-blade geometry and an operational load scenario, as described in Section 1.4 and Section 2.3.

The intention is that the designs and results from such a generalised investigation will provide new quantitative insights into possible structural buildups in advanced modern propeller-blade geometries. Such blades have a considerably more complex coupling of the geometric features of the blades and the composition of materials comprising the blade. This paper examines the following four plausible ideas for designing desirable blade deformation characteristics and how they compare: hollowing the blade, anisotropic surface laminates (with and without local reinforcement patches), internal structures and freeing the tail of the propeller blade. These concepts are described in Section 2.2.2. Note that previous studies were basically limited to anisotropic surface laminates of polymer-based composites.

The finite element analysis (FEA) used in this paper is performed with the FEA software Abaqus 2017 by Systema Dassaults [36]. An Abaqus FEA methodology is applied based on FEA methods that have been explored and experimentally verified in previous publications [8,27,28,29,37,38]. The paper that explored the experimental verification of the FEA method on a composite propeller was previously published in *Polymers* [38]. With this starting point, it is assumed that it is reasonable to investigate design concepts numerically and thus identify promising design concepts. Nevertheless, straightforward manufacturability might not always be a core requirement in numerical models exploring the structural elastic response of simplified material compositions. 

The previous study and this study both focus on the same propeller-blade geometry. The FEA methods applied in this design have undergone mesh sensitivity and element selection studies and have been used in an experimental verification of a manufactured composite prototype [27,28,29,38]. Therefore, the FEA methodology applied in this paper is grounded in reality. Details of the FEA methodology are given in Section 2.2 and Section 2.3. The element types used in the FEA simulations are given in Section 2.2.1 and Section 2.3.2. Several different FEA element types are applied to model the same designs to explore the influence of choosing different elements, as various modelling methods can provide slightly different results depending on the FEA element definition and interactions.

In addition, any propeller-blade design that is intended to be commercially reliable requires a product development process to explore manufacturing methods and processing as well as further insights through other disciplines. In addition, the simulations in this paper and new simulations require experimental verification of the design’s elastic deformation characteristics, estimated intrinsic properties, and performance. As such test campaigns also require a prototype development process and are extremely time-consuming and expensive, it was considered out of the scope of this initial adaptive propeller-blade design process into high bend–twist coupling propeller blades.

### 1.4. Exploring Bend–Twist Design Concepts on a Generalized Blade Structure

A short flat symmetric foil-blade model was made in Abaqus 2017 as the generalized blade to quantitatively investigate the proposed design concepts. The NACA0009 blade model is shown in Figure 1.

The chosen NACA blade beam has a chord line length of 1 m and is extruded to a 1 m span: basically, a short, wide fixed beam in an airfoil shape. With its nonvarying cross-section, flat shape and square frame, the geometry resembles a blade from Hooke’s screw propeller from 1683; Carlton shows an illustration of this in his book [2]. The NACA0009 blade is a simple propeller-like foil structure made to explore the effect of varying the structural buildup of a blade with limited simple geometric features.

To be able to observe blade models’ elastic responses and how their deformation characteristics vary based on various material compositions, a load case is needed. The load case chosen for the NACA0009 blade was a simple uniform pressure on one side and suction on the other. These are significant simplifications compared with commercial marine propeller-blade operational loads. However, this approach was deemed sufficient, as the interesting result is the changes in the deformation characteristics between the different structural buildups, not the specific deformation characteristics from the load itself.

The specifics of the quantitative investigation into the separate design concepts on the generalized blade are given in Section 2.2.

### 1.5. Modelling and Dimensioning a Typical Modern Bend–Twist Propeller Blade

After exploring the design concepts of the generalized NACA model, the concepts are applied to the bend–twist propeller-blade design for a typical metal propeller-blade geometry. This propeller blade is a commercial metal-blade design that has been used in the published literature [8,27,28,37]. The blade design has several unique propeller geometry features, such as a significant blade skew, a varying sectional foil profile along the blades span and a propeller radius of 650 mm. The goal is to design specific deformation characteristics in the propeller-blade design for one particular periodic load-variation load case using the investigated design concepts.

The deformation characteristics in all blade models are evaluated by tracking the foil-shaped cross-sections throughout the blade span during loading [8,27,28,29]. The parameters tracked in this study were blade deflection, blade twist, bend–twist (coefficient of twist per bending deflection) and camber (foil shape indicator). These parameters were tracked at several blade radii and plotted. As the blade was loaded, the change in these deformation characteristic parameters allowed for evaluating whether a blade design had sufficient bend–twist coupling.

As adaptive bend–twist propellers use deflection to obtain the desired twisting, deflection is a requirement. In and of itself, blade deflection is not much of an issue, but it must be accounted for when making the propeller-blade shape by considering both the unloaded shape just after production and how the blade will be shaped during operational cruising loads. In addition, the design should consider a pitching change in the adaptive composite propeller. Therefore, the desired design criterion is that the bend–twist parameter in the blade should be maximized to ensure an efficient twisting pitch change. By first maximizing the desired deformation coupling, later, when designs move towards commercial designs with specific intended deformation characteristics for expected operational scenarios with load variations, the bend–twist effect can be tuned down if an overcompensating twist is found in the design [10,35].

For the final propeller-blade design, the goal was to obtain a combination of as much bend–twist as possible while aiming for a specific deformation characteristic, i.e., a desirable twist causing sufficient blade pitch change between the minimally and maximally loaded blade in a specific periodic-load variation. The periodic-load case investigated in this paper has been used in previous studies [8,27,28]. It has been shown that when the thruster is rotated to an 8° steering angle relative to the vessel’s travel direction to change the heading, the periodic-load variation almost doubles the total blade load between the minimum and maximum load [8,27,35]. Earlier works have also identified the angle change in the propeller-blade inflow between the minimum and maximum load cases to be 2.3° at half the blade length, 1.3° at 70% of the blade length and 0.8° at 90% of the 650 mm radius propeller blade [8,27,28]. It is assumed that if the pitch twist in the propeller blade between the minimum and maximum loads is of similar magnitude, the propeller-blade design will mitigate the propeller’s inflow changes, potentially mitigating the periodic-load variation.

A more considerable twist change was needed closer to the blade’s root, and a smaller twist change was required further towards the tip, which was attributed to the rotational velocity increasing at larger radii. The fact that a larger twist change was required at lower blade lengths complicated the design challenge, as propeller blades usually deflect the most at the tip, and if the twist is related to the deflection, the blades will twist the most at the tip. Furthermore, in previous work on bend–twist propeller blades, the twist has usually been largest at the blade’s tip [5,8,13,27,28,37,38]. Therefore, trying to ease the twisting at the blade’s tip in the final propeller-blade design is suggested to not overcompensate for the inflow variations at larger radii.

The paper’s structure is as follows. First, the material selection is defined before the modelling and investigation details are described. Then, the details regarding the numerical simulations of the NACA blade are provided with a suggested design that combines all design concepts. Afterwards, the findings are applied to design a propeller blade with desirable deformation characteristics and sufficient static strength for the typical propeller-blade geometry. Finally, a discussion of the study is provided before the conclusions are drawn.

## 2. Materials and Methods

This section describes the materials used in the numeric models and the specifics regarding the FEA for the NACA blade and propeller-blade models.

### 2.1. Materials

Traditional solid rigid metal propellers are often made of nickel–aluminium-bronze alloys due to the noncorrosive properties of the metal [39]. Nickel–aluminium-bronze has a Young’s modulus of approximately 115 GPa.

Unconventional deforming propeller-blade designs require unconventional propeller materials that can show large deformations. In addition to composites of anisotropic fibre-reinforced polymers (FRPs), isotropic materials such as metals and polymers are considered in this paper for adaptive propeller blades. An advantage of numerical modelling is that materials with relatively different properties, such as stiffness and strength, can be tested quickly. Furthermore, the wide range of plausible high-deformation materials can help fine-tune the total blade stiffness and deflection, which could be useful when iterating on propeller-blade designs to approach a particular set of deformation characteristics under a load case. 

This paper applies carbon fibre-reinforced epoxy/plastic (CFRP) and glass fibre-reinforced plastic (GFRP) FRP materials, the most common materials used in adaptive polymer-based composite propellers [39]. For FRPs, the anisotropic ratio between the stiffness coefficient in the fibre direction and the stiffness coefficient across the fibre direction when the FRP is in the UD configuration indicates the potential for bend–twist deformation [7,8]. For example, while GFRP usually has a ratio of approximately 3–8 between these stiffness coefficients, CFRP has a ratio of approximately 8–13, indicating that CFRP is the more favourable material for bend–twist coupling. Ideally, a very high modulus fibre combined with a very compliant matrix should be used to obtain maximum anisotropy of the composite. 

The XPREG product series from EasyComposites Ltd., Stoke-on-Trent, UK, a prepreg system with woven fibres and unidirectional fibres, was chosen for the CFRP. The material properties of the CFRP were experimentally measured in previous research studies [27,28,29]. Regarding the GFRP, a set of published material properties was used [40]. The material properties are given in Table 1.

The isotropic materials of maraging steel (grade 250), titanium (alpha–beta alloy), aluminium (6061 alloy) and epoxy plastic give a wide range of stiffnesses and are considered in this study. The respective Young’s moduli for these isotropic materials are 190, 120, 70 and 3 GPa, and the Poisson’s ratios are set to 0.3 for all materials [41,42,43]. While the plastic is very soft and weak compared to the other materials, it might be helpful as a substructure that experiences relatively less load, just as balsa wood has been applied in combination with FRPs and other stiff materials in wind-turbine blades.

#### Strength Estimation through Failure Criteria

Failure criteria are used to evaluate the material stress state under loading to assess the realistic feasibility of the suggested designs. Some factors must be taken into consideration when designing propellers consisting of metal and composite subcomponents comprising the blade structure. Ideally, material samples of commodity materials should be experimentally tested, but material data from online material property databases were applied for simplicity.

First, all constituent materials in an adaptive propeller-blade design should only be stressed in their elastic range in elastic adaptive propeller blades to avoid failure or permanent deformation to the propeller-blade geometry. Isotropic metals are then limited by their yielding stress. To add an engineering rule-of-thumb safety factor, 80% of metal yield stress is taken as the maximum stress limit for the metals (80% of 655 MPa for maraging steel, 1000 MPa for titanium and 270 MPa for 6061 aluminium) [41,42,43]. For reference, a nickel–aluminium-bronze alloy usually exhibits a yield stress of approximately 470 MPa. The epoxy had an ultimate tensile strength of approximately 60 MPa [44], but a safety factor of 2 was applied to prevent wear. A polymer that can take higher strains to failure, such as some nylons, might be more favourable for future designs, as deformations in adaptive composite blades can be large.

For the FRPs, a maximum strain limit of 0.5% was chosen as the failure criterion. This strain level is a rule-of-thumb strain limit against matrix cracking in the FRP. For simplicity, when evaluating the strengths of a model, the stress-based (80% of yield stress) strength criteria of the isotropic materials are converted to maximum strain criteria: 0.25% for maraging steel, 0.8% for alpha–beta titanium, 0.3% for 6061 aluminium and 1% for epoxy. More accurate evaluations need to be made for real designs, but the conceptual approach developed here should remain valid.

On a side note, it should be mentioned that material property values for stiffness and strength are based on room-temperature data, while a completely different temperature profile might be present in operational conditions, altering these properties slightly [45]. Environmental effects, abrasion and erosion resistance, material ageing and fatigue under a periodic-load variation should also be evaluated for sufficient durability before moving towards commercial designs [3].

### 2.2. Simplified Blade NACA0009

Three types of NACA blade models (described in Section 1.4) were made: the baseline model, exploratory design models and combined design model. The baseline is used as a reference, the exploratory models are design examples with different design concepts, and the combined designs apply all design concepts to obtain a plausible design with good twist.

#### 2.2.1. Setting up the Simulation for the NACA0009 Blade

In Abaqus, the NACA blade model was assigned a multipoint constraint (MPC) to connect all degrees of freedom of the nodes on the blade root of the model to an Abaqus reference point. Then, a boundary condition (BC) was placed on the reference point, fixing all degrees of freedom on all the nodes on the blade root to resemble a rigid connection fixture for the NACA blade. The MPC and BC are indicated in yellow in Figure 2. The boundary condition is designed to be comparable to a fixed connection to the hub, for example, through geometric fitting, light clamping and possibly bolts. Designing the details of the connections was not necessary for investigating the principal effects.

The load on the NACA blade was applied as 2500 kg of evenly distributed pressure on the top side of the blade, and the same load as a negative pressure was applied on the bottom side of the blade. The load magnitude was set based on initial studies on the NACA blade models, as described in Section 2.2.2.

The model meshing was performed with an element size set to 1 cm. Different element types were used for the mesh in the different models. For all solid structures in the paper, element type C3D10 was used. For the continuum-shell elements, STRI65 was used because it is the standard option when applying such elements in combination with the Abaqus composite layup manager and material manager, which were used to apply the material definitions in the blade models. For hollow-shell models, element type S8R was used for the shell elements.

#### 2.2.2. Setting up the Simulation for the NACA0009 Blade

With the materials and simulations defined, the material layup compositions and deformation characteristics are free to be investigated. A solid isotropic baseline model functioned as a reference for comparison, inspired by [13]. The baseline model is a solid FEA grid of a solid aluminium model.

The NACA blade is a simplified structure to provide insight into what general deformation characteristics various material compositions and design approaches cause. This will help to design more realistic blades later. By quantitatively comparing the deformation characteristics of the design choices based on the different design concepts, a generalized indication of the design concepts’ bend–twist potential and possibly promising design choices that cause desirable traits can be identified. The simulations into the considered design concepts are used to predict traits in the deformation characteristics when applying the design concepts. Four design concepts were considered in this paper and are listed below.

Design concepts:Hollowing a solid propeller-blade geometry;Hollow composite blades with anisotropic material bend–twist coupling;Internal or segmental reinforcing substructures in mostly hollow isotropic blades;Blades with a root fixture at the leading edge of the blade but a free tail/trailing edge.

Models based on these design concepts are illustrated in Figure 3 and described in this section. The manufacturability of these design concepts and the final combined NACA model are not considered in detail. Metal substructures could be achieved with additive manufacturing methods in all considered metals, and composite and plastic manufacturing methods are highly versatile when arranging fibres for specific reinforcement. Joining of substructures (geometrically, mechanically or adhesively chemically) is not considered. However, a prototype development process mapping these aspects is required for possible commercialization and experimental verification of the bend–twist blade model.

Three discrete design choices within each design concept were made, but the development process is not shown. However, with so few models within each design concept, the investigation in this paper can only indicate what each design concept can contribute to the desired elastic response, and approaching an optimised solution for each design concept is not feasible in this work. The models of the design choices investigated are provided in the results section to keep the models close to the corresponding deformation characteristics.

The design concept of hollowing the blade is a natural consequence of going from a solid metal propeller to a composite propeller, as composites are generally made as laminate shells. It is explored as a design concept because, depending on the blade geometry, simply hollowing the blade could somewhat surprisingly cause bend–twist characteristics in a blade. It was deemed relevant to separate the material and geometric contributions to the subtle deformation characteristic parameters before later combining them to observe each disparate contribution and their coupling. Three NACA blade models with a 1 cm thick aluminium shell were made to explore the change in the deformation characteristics from hollowing this geometry. These models in these FEAs were used as the basis when further developing FEAs for the design concepts: anisotropic composite layups and reinforcing structures. The models were modelled with different methods as it was suspected that the shell elements might have issues due to the emphasis on the thin-shell behaviour and overlapping material volume in curved sections, while solid-element-based hollow models might not sufficiently register the thin-shelled behaviour. 

The second concept, the anisotropic surface laminate propeller blade, has been extensively applied in previous research and has been used to make functioning bend–twist propeller blades using this concept alone [5,8,12,13,19,20,27,28,29]. In addition, some studies have generated laminate layups with genetic algorithms using FRP stacking arrangements in which the propeller blade is within the given parameter [5,20] but only within the considered solution space. Others have applied novel new ideas, such as curved fibre paths in the anisotropic shells [46], showing that there are still more possibilities for improvement in this design concept alone.

The common method for bend–twist deformation coupling in layups is to utilize plies of straight fibres orientated at certain angles to contribute to the coupling coefficient in a structure. Prior studies have shown that arranging the fibres to point 30° off-axis to the bending direction, slightly towards the leading edge, seems to be the best for unidirectional contribution to twisting in anisotropic laminate layups. For woven or biaxial 0°/90° plies, 22.5° appears to be the optimum because the perpendicular ply slightly opposes the desired twist contribution [8,9,13,16,27,28,29].

In addition to the bend–twist contribution of plies oriented at approximately 22–30° off-axis, a laminate must have sufficient fibres in other orientations to ensure sufficient strength in all necessary material directions to be feasible as a structural material. However, if the plies contributing to strength have lower stiffness than the bend–twist plies (using GFRP for strength instead of more CFRP), the bend–twist characteristics of the laminate are better than if the strength plies have equal or higher stiffness coefficients [8].

Local quasi-isotropic reinforcement patches were added to the laminate. Such a vertical flange around the maximum foil thickness, which is commonly used in wind-turbine blade layups, can enhance the bend–twist properties [8,27,28,29].

The first of the last two design concepts investigated structural reinforcement close to the leading edge [8]. Reinforcing structures in foil designs is a known concept, but they are still not known to be applied to adaptive composite marine propeller blades. Reinforcing substructures can have anisotropic material properties, as Ong and Tsai did with their composite D-spar for wind turbines with bend–twist properties [7]. However, our research scope focused only on internal structures with solid isotropic material properties, as described in Section 2.1.

Freeing the tail is a relatively unconventional new design concept supported by the logic that a relatively stiffer leading edge on the blade could also mean a less restrained trailing edge. The concept is, to some extent, inspired by bird wings and aquatic fins and their skeletal structure. A practical way to imagine this concept is to pretend that one can saw, rout or mill a grove from the trailing edge along the root connection, leaving only a connection in the blade’s leading edge. For marine propellers, traditionally, a sturdy connection between the blade and the hub is good in rigid propellers, but structurally freeing the tail might be better for adaptive propellers, as it leaves the trailing edge less constrained. 

##### Combined NACA Blade

After the deformation characteristics for the baseline and exploratory models have been mapped, a combined design for the NACA blade is proposed through an iterative approach to merge promising design choices. The exploratory design concept models cannot simply be superpositioned as a solution, as the resultant design would be too stiff to show any significant desirable deformation and probable volumetric overlap of materials.

A new design is developed with two main aims: that the design is made rigid enough not to strain any constituent material over the failure criteria set in Section 2.1 while also being compliant enough to flex under the expected loads.

### 2.3. Designing a Combined Material Layup for a Bend–Twist Propeller Blade

After the NACA blade investigations, a typical propeller-blade geometry was next. The propeller geometry was a commercial metal propeller-blade geometry. Ultimately, the goal was to propose a bend–twist propeller blade that showed a desirable twist between the minimum and maximum load in a periodic-load case. The blade should also have sufficient strength based on the set failure criteria.

While the propeller-blade design should be based on the concepts of the NACA blade investigation, the NACA blade and the investigated propeller blade differ geometrically, as shown in Figure 4. The difference in the geometries might mean that the design concepts do not transfer directly. Therefore, transformation alterations should be made to the NACA design choices within the design concepts to find good propeller-blade design choices. Figure 4 shows how the blades differ with the rounded frame of the propeller blades and a nearly 30° skew to the right, indicated by the black lines and the dotted arrow in the figure. The propeller blade’s foil-shaped cross-section also varies in shape characteristics and pitch throughout the length of the blade.

#### 2.3.1. Setting up the Simulation for the NACA0009 Blade

The loads on the blade depend on the fluid passing along the blade. This is called fluid–structure interaction (FSI). The effect can be modelled simply as a one-way FSI, i.e., only the initial loads of the fluid on the undeformed structure are considered. It is assumed that changes in the shape of the blade will not affect the loads. A two-way or higher-order FSI would consider the interdependency between the fluid flow and loads on the deformed structure. For example, to quantify how well a bend–twist propeller-blade design mitigates forces in a load variation, a two-way FSI is needed. However, to indicate if the elastic response of a blade model initially causes much twisting, which has been shown to cause load mitigation, a one-way FSI is sufficient.

Kumar and Wurm found that the difference in elastic response and deformation characteristics between a one-way FSI and a steady-state two-way FSI for their adaptive composite propeller-blade design was that the two-way FSI scaled the deflection of the tip down by almost 14% and the blade thrust down by nearly 7% [5]. However, if a blade design has more remarkable deformation characteristics, the dissipation of the deformation might be even more favourable and give a more considerable thrust dissipation [10].

The fluid loads were applied as a pressure distribution on the blade surface (neglecting the shear forces from the fluid), derived with a one-way FSI provided by KONGSBERG Maritime Ulsteinvik [35]. The operational-load case of FSI was modelled for a propeller on a thruster rotated by 8 degrees to alter the ship’s heading. The vessel was travelling at 25 knots with a propeller rotation of 561 RPM [35]. The centripetal forces of the propeller rotation were taken into account. The instantaneous blade loads on all five blades are shown in Figure 5a, and the blade thrust contribution is shown in red. Two numeric simulations modelling the maximum and minimum load loads were conducted. The magnitude of the resultant load almost doubled in the maximum-loaded case compared to the minimum-loaded case (in this one-way FSI analysis). Another difference was that the pressure distribution characteristics varied between the minimum and maximum cases, as shown in Figure 5b,c.

As a result, the two pressure distributions had different resultant force directions and different locations of the resultant load on the blade (centre of pressure) [8]. The blade thrust (the red line in Figure 5a) shows that the thrust is 15.2 kN for the least-loaded case and 29.4 kN for the most-loaded case, meaning that the load variations were 14.2 kN.

#### 2.3.2. Propeller-Blade Models

To simulate the deformation of the blade, an appropriate FEA was set up for the propeller blade. First, the BC was set up in the same manner as for the NACA blade, as shown in Figure 2.

Again, a solid aluminium blade case was made as a baseline model. This time, only the blade geometry-dependent design concept of hollowing the blade was investigated before combining all the design concepts in a combined design. The hollow propeller blade is useful for comparing with the NACA blade to observe the geometric effect and load characteristics contributing to the bend–twist before adding any further design concepts. The final combined propeller-blade design can then be compared with the hollow isotropic design to isolate the bend–twist contribution due to the material composition.

The same elements were used for the propeller-blade models as the NACA blade models in Section 2.2; C3D10 elements were used for the solid isotropic components and STRI65 elements were used for the anisotropic surface laminates according to the experimentally verified FEA methodology used in this paper. The element types are also the Abaqus standard for isotropic and anisotropic materials respectively. The element size was set to 3 mm, as an earlier study on this blade geometry found that displacement results converge for elements of this size in a mesh-sensitivity study [27,28].

While the NACA blade uses straight cross-sections in Figure 2, considering foils as if the blade travelled straight forward, the propellers travel forward while rotating around the propeller axis. The propeller rotation causes the water to follow a circular path with a constant radius over the blade, as shown in Figure 6. The radial lines of 0.5, 0.55, 0.6, etc. up to 0.95 and 1.0, some of which are shown in Figure 6, were tracked on the propeller-blade model to plot the observed deformation characteristics along the blade span with high fidelity.

## 3. Results

This section gives the results of the NACA blades and the propeller-blade FEAs. First, the NACA blade models with their respective deformation characteristics are shown before the NACA blade investigation is summarized by comparing the deformation characteristics of the simulation. Finally, the design and deformation characteristics of the propeller-blade geometry are given. 

The desired characteristic is maximum twisting per bending without changing the hydrodynamic shape of the blade. How this can be achieved is first described for the simple NACA blade to show the principal approach. The second part shows that the same principles also apply to a realistically shaped propeller blade.

### 3.1. NACA Blade Models

Fourteen models with different designs were made for the NACA model: one baseline model, a simple solid metal blade, twelve exploratory design models and one final design for the NACA blade model. The manufacturability is not primarily considered in these models, so product development work into assembly strategies needs to be carried out for experimental prototyping. Nevertheless, as the global stiffness, elastic response, and deformation characteristics are still expected to be present, the method is deemed adequate. The exploratory models test the structural stiffness response of different discrete material compositions and the subsequent deformation characteristics and their relative quantities. The combined design is intended to illustrate roughly how an adaptive propeller’s structural composition could look.

#### 3.1.1. Hollow NACA Models

The first model in this concept was a hollow solid, as it used solid elements to model a hollow blade. The second hollow-blade model used continuum-shell elements, while the last hollow model used shell elements.

The displacements in the hollow NACA models and the solid baseline case are shown in Figure 7. 

Plots (a), (b) and (c) in Figure 7 all model the same hollow blade with a 1 cm thick aluminium shell but are modelled with various element types to see the range of FE predictions. They show a similar displacement contour; however, none fully agree. Specifically, (a) and (b) show thin-shell behaviour at the surface of the blade caves, while (c) primarily shows thick-shell behaviour. In addition, it seems that (b) and (c) capture some geometric stiffness effects where the blade models’ sides join along the tail/trailing edge. The solid reference model is shown in (d), which shows less deformation. While the displacement contour plots in Figure 7 are sufficient to give a global overview of the blade deformation, it is difficult to determine the models’ more subtle deformation characteristics mentioned in Section 1.2. Therefore, the cross-section foil profiles in Figure 2 were applied. The foil profiles at 50%, 70%, and 90% of the blade length were plotted for the unloaded and loaded cases for the four baseline models, as shown in Figure 8.

While all the required information is visually present in the deformation contour and foil plots in Figure 7 and Figure 8, a more straightforward quantitative visualization was required to easily compare the models. Therefore, quantifying the deformation characteristics in the baseline model and the hollow models along the blade span was conducted with the method in [8]. The results are shown in Figure 9, which shows how the blade changes with increasing radius (x-axis) for different configurations. The changes are as follows:Blade deflection;Pitch angle α of the chord line.Pitch angle α of the chord line normalized by the deflection.Change in the camber.

A thorough explanation of the application of the foil-deformation theory method, which is used to extend the results shown in the general displacement plots in Figure 7 to the plots in Figure 8 and Figure 9, is described in Ref. [8]. In short, all nodes on the lines in Figure 2 and Figure 6 are tracked in the simulations to quantitatively and unambiguously determine the blade foil deformation. In Figure 9, the plot on the left shows the average deflection of all nodes on the radial foils along the blade span. The next plot on the right shows the absolute twist in the blade foils. The third plot indicates where along the blade an efficient bend–twist effect is occurring, while the fourth plot indicates whether the foil is changing shape.

In Figure 9, the solid baseline model can be seen to deflect approximately 3.5 mm at the tip and twist −0.05°, which is the desired direction. The three hollow-blade models exhibit similar but slightly different characteristics, with a deflection between 9 and 10.5 mm and a twist between −0.1° and −0.2°. The shell model does not model the interconnectivity between the shells at the trailing edge, meaning that the accurate solution might lie between the continuum-shell model and the hollow-solid model. The continuum-shell model acknowledges that the two shells conjoin at the blade’s trailing edge but does not consider that through-thickness shear occurs. On the other hand, the solid elements might not be captured if the propeller sides show thin-shell characteristics.

#### 3.1.2. Anisotropic FRP Layup

While the hollow model based on shell elements had a weakness by not considering the interconnectivity of the shells on the trailing edge, it has a significant advantage when modelling various anisotropic layups. When using models based on shell elements, the FEA software automatically generates the material volume, which is a faster process that requires less modelling work compared with manually modelling the physical material volume using continuum-shell elements or solid elements before defining materials. However, as mentioned in Section 3.1.1, this method does not consider the effect of the propeller side-material volume joining at the trailing edge.

Three models with CFRP skins were made to investigate the effect of different layups. The first model was made with a standard quasi-isotropic CFRP layup, with only all-over plies in the 0° and 45° directions and no local reinforcement patches. This model was used to gauge how fibre reinforcement was required for the chosen load case before any bend–twist tailoring was attempted, i.e., a simple quasi-isotropic blade.

Then, a model with a layup that used many more CFRP UD plies in the 30° orientation and CFRP woven plies in the 22° orientation was made. These angles were chosen for the orientation because they maximize the bend–twist coefficient contribution of the plies, as mentioned in Section 2.2. In addition, some additional woven fabrics in 0° and 70° plies were added to reinforce the laminate to ensure the model’s strength in these fibre directions. The coordinate system for fibre orientation onto the propeller is given in Figure 4.

The final model, based on the anisotropic FRP layup design concept, used the layup in the previous model and added a reinforcement patch resembling a flange, as shown in Figure 10. It is expected that the flange, while adding global blade stiffness and lessening the deformation and, thus, twist, might cause a higher bend–twist coefficient in the model. This flange consisted of some extra UD plies in the 30° orientation and woven plies in the 0°, 20° and 70° orientations. The layup orientation of the flange was iterated manually with the angles for bend–twist contribution and material strength. Illustrations of the models are shown in Figure 10, and the layups of the three models are given in Table 2.

The deformation characteristics are compared in Figure 11. The second model twists the most, while the third model (adding a flange) shows better bend–twist characteristics.

As expected, based on the numerous publications on the design concept, a twisting deformation and significantly improved bend–twist coupling were achieved using the anisotropic layups in Figure 11 compared with hollowing the NACA blade in Figure 9. The composite designs are also seen to deflect more than the aluminium used for the baseline and hollow NACA models. While it is known that the design concepts contribute to bend–twist coupling, given the short span feature of marine propeller blades, it is of particular interest to compare the bend–twist efficiency and twist magnitude with the other design concepts in the following sections, as mentioned in Section 1. Note that the models in this design concept were based on shell elements, which were shown to slightly overestimate both the deflection and twist, as seen in Figure 9. However, even when considering this overestimation, the anisotropic design models showed improved bend–twist coupling.

#### 3.1.3. Internal Structures

The investigation then proceeded to the design concept of internal reinforcing structures. The explored structures were simple solid isotropic geometries that fit inside the blade under the surface shell towards the blade’s leading edge. Regarding the manufacturing and assembly, these titanium substructures would require some geometrical mechanical, welding-based or adhesive fastening to the outer aluminium shell in a product development process. However, in these simulations, they are fixed with the Abaqus TIE constraint feature. An additional BC was added to the bottom surface of the internal structure, fixing it at the root, because such an internal structure would probably extend down into the blade’s fixture and be mechanically fixed.

For the simulations, the internal structure model was added to the NACA continuum-shell hollow model, and the Abaqus TIE constraint was used to fix the internal structure surface to the inside surface of the hollow blade shell. The leading-edge mast and cornerstone were the first two internal structure models in the design concepts’ design series. The final internal structures model used another variation of the leading-edge cornerstone, referred to as the framework design. The models are shown in Figure 12.

The material choice and geometric features were not optimised to the limit of the material properties but were dimensioned for the designs to show comparable deflection based on the two previous concepts. The deformation characteristics of the internal structure models are compared in Figure 13. The second model twists the most and shows the best bend–twist characteristics. While all design models having internal support structures show improved bend–twist properties, the final twist is smaller than for the anisotropic layups, possibly due to the smaller deflection.

Another type of interesting internal structure is ribs. While ribs are expected to be vital to maintain the propeller-blade shape when loaded, they were not expected to cause bend–twist deformation and were therefore not investigated as a design concept or design choice that causes bend–twist properties. Nevertheless, rib structures were included in the combined design models for both the NACA blade and the propeller blade to limit the change in the blade’s foil shape.

#### 3.1.4. Freeing the Tail

Finally, the idea of freeing the propeller blades’ tail to allow for more twist was explored. The first model explored the effect of freeing the tail by releasing the nodes in the back half of the blade from the BC. The second model released two-thirds of the nodes from the BC, while the third released 75% of the nodes. It is almost as if one is cutting more and more into the blade root from the tail in each model. The change in the MPC and BC to free the tail for the three exploratory models is shown in Figure 14.

The deformation characteristics of the models with the different levels of free tail are compared in Figure 15. It can be seen that the more the tail is loosened, the more twist is obtained. At the same time, the difference in the bend–twist characteristics between the models is small. It changes little between releasing 66% or 75% of the tail. The extra deflection in the model by releasing 75% leads to an extra twist in this model. The camber of the blade is only slightly affected by releasing the tail of a solid blade.

#### 3.1.5. Combining All Bend–Twist Design Concepts in a NACA Design

A final design was proposed that consisted of a hollow anisotropic surface laminate with reinforcing structures and a freed tail. The design was composed of parts inspired by the design choices from the exploratory models. However, the design concept models could not be directly applied to ensure that the new model was not too stiff to deform, as adding several components together increased the model’s overall stiffness. For example, while the layup from the third design in the anisotropic layup design series was used, a backing ply was removed from the layup to increase deformation. In addition, two ribs were added to help maintain the blade’s profile. An overview of the buildup of the final NACA blade is given in Figure 16. For this design, a manually iterative exploratory design process to push the constituent materials’ strain closer to their limits for more bend–twist coupling was applied, resulting in a large deformation in the blade. The FRP layups in the ribs and surface laminates are given in Table 3.

The deformation of the final blade, the foil cross-section plots and the deformation characteristics plots comparing the design to the solid baseline case are given in Figure 17.

The deformation characteristics show that the design using a combination of desirable design concepts outperforms all other designs that only employ one design concept. The design also deflects much more than the other designs, pushing the design close to the material failure limits. The strain values in the CFRP laminate and rib are maximized at 0.492% with regard to strength. The steel part is not strained to a level that triggers the failure criteria with a maximum strain of 0.247%. The strain in the plastic part of the reinforcement structure is maximized at approximately 0.86%, indicating that the static strength of the final combined design composition is sufficient. Some single-node, high-value stress concentrations are present, but these are deemed artificial from the modelling method and neglected in these simulations and subsequent results.

#### 3.1.6. NACA Blade Investigation Summary

As only three design choices for a bend–twist design were investigated within each design concept, the focus was on showing plausible bend–twist design models within each design concept and quantitatively examining the bend–twist design concepts’ expected contribution to deformation when applying these concepts. An overview of the relative bend–twist efficiencies of the NACA blade models is shown in Table 4. The absolute values used to calculate the relative values in Table 4 are given in Figure 8, Figure 9, Figure 11, Figure 13, Figure 15 and Figure 17.

Table 4 shows that all the exploratory designs had 2–20 times more twisting than the reference baseline. Looking at the twist per deflection characteristic, the bend–twist efficiency of each design can be determined. For example, the hollow isotropic design concept does not significantly increase the twist per deflection (bend–twist) for the NACA blade, while the best anisotropic designs more than double the twist per deflection.

Both reinforcing structures and freeing the tail improved the bend–twist of the blade. Reinforcing structures showed great bend–twist characteristics over the whole blade, but due to the extra stiffness in the blade from the structure, the blade twist was not as pronounced as in the anisotropic laminate design concept because the blades’ deflection was limited. An exclusive feature in the design concept of freeing the tail was that these design models showed good bend–twist coupling at the blade’s smaller radii.

The combined NACA blade design that applied all the considered design concepts achieved excellent bend–twist coupling. The final design deflected more than any other NACA blade design, but since the constituent material failure criteria were not triggered, the large deflection was acceptable. As a result of the large deflection and the bend–twist property, a good twist was achieved. With the achieved deformations, it seems likely that the design concepts synergize. It was observed that the design concepts of reinforcement structures and freeing the tail seem to synergize, as the reinforcement keeps the leading edge in place, while freeing the tail allows the tail to deflect, causing a twist to manifest down the blade. It seems, however, that the camber of the blade’s shape is affected by this reinforcing structure/freeing the tail coupling, as the final design shows some camber change.

### 3.2. Typical Propeller-Blade Design

Next, these design concepts were explored in terms of the propeller-blade geometry. A solid metal propeller-blade reference was investigated first, followed by modelling of the hollow blade. The hollow-blade models are also reference models intended to isolate and identify the bend–twist contribution of the unique geometric features of the hollow-propeller geometry. Subsequently, a composite-blade layup design was investigated by combining the hollow blade with anisotropic surface laminates, reinforcement flanges, inner reinforcement structures, and a modified free-tail root connection. By performing these step by step, the geometric contribution and design choice contribution to the bend–twist properties can be observed more distinctly.

Because the propeller-blade model was smaller than the NACA, a 5 mm thick aluminium shell was found to be sufficiently stiff and strong under the load case, while still flexing, for modelling the hollow-blade models. When performing the investigation, the commercial propeller-blade CAD geometry surfaces proved to be complex, making it too difficult to make a mesh that allowed for a continuum-shell element model-based analysis. Therefore, only two hollow cases were made: solid hollow and shell. These models were simulated with the maximum periodic-load case. The deformation characteristics are shown in Figure 18.

When examining the blade deformation characteristics of the solid propeller baseline, it can be seen that the blade twists the wrong way. On the other hand, the hollow models already show an approximately 1° twist towards the desired direction, which means that the chosen propeller-blade geometry naturally shows favourable bend–twist coupling when hollowed out. In Figure 18, the solid and hollow curves for bend–twist coupling follow similar patterns but with an offset. It was decided to use the hollow-solid model as the reference baseline model for the following combined propeller-blade design because the solid baseline model twists the wrong way.

#### 3.2.1. Applying the Design Concepts to a Commercial Propeller-Blade Geometry

When designing the blade composition, the maximum periodic-load case was applied to the propeller-blade geometry. First, the combined blade design was used as inspiration for a starting point. Then, after considering the intermediary starting point design that showed a large bend–twist coupling and twist, the second design criteria are introduced. The design should show a significant twist, but the twisting at the blade’s tip should be reduced for the reasons mentioned in Section 1.2. It was found that changing the angular orientations of the bend–twist-contributing plies away from the optimum bend–twist angle was a quick solution to tailor the bend–twist deformation characteristics towards easing off the tip. The design was finally evaluated for its static strength with small final fixes to avoid overshooting the failure criteria.

The final design suggestion is shown in Figure 19a. The blade design has a solid internal reinforcing structure of maraging steel and an FRP rib structure. In addition, an anisotropic shell was used with a reinforcing patch on the suction side. The FRP layups are given in Table 5. The final design choice was to release 70% of the tail connection. The remaining BC connection is coloured yellow in Figure 19a.

High local strain values are visible along the leading edge and on the skin in the metal substructure plot in Figure 19b. These strain concentrations are most likely artificially caused by the sharp angle of the metal edge into the triple connection point between the two anisotropic shell elements and the internal structure and the infinitely rigid surface-to-surface Abaqus TIE constraints. The strain of the bulk steel was generally quite low. However, one practical modification was made to the substructure after evaluating the design for strength: the metal substructure tip was cut off where it showed a thickness of 4 mm. The alteration was necessary because the metal showed an extensive bulk strain in the sharp tip. The design change did not affect the modelled global blade characteristics.

Regarding the manufacturing and assembly, some geometrical, mechanical, adhesive, hybrid bonding or insert moulding modifications are needed to attach the substructures to the composite skins. However, that product prototype development was out of our scope and budget. The final design shows some desirable twisting characteristics and sufficient strength under maximum periodic load. The design was then subjected to the minimum periodic load to examine the twist difference in the blade between the cases and ensure sufficient strength under both load cases. The pitch change in the design between the minimum and maximum loads is compared to the angle change in the fluid inflow (introduced in Section 1.2) in Section 3.2.2.

The laminate layup of the FRP In the rib and surface laminates of the blade design is given in Table 5. Regarding the blade’s strength, the strain contour of the internal shows that, apart from seemingly artificial stress concentration geometry that seems numerically based, the blade seems to have sufficient strength for starting to approach more detailed prototyping designs.

The global displacement contour of the blade under maximum load is shown in Figure 20a, and the deformation characteristics of the blade under maximum and minimum loads are shown in Figure 20b. The deformation characteristics are compared with the baseline case in Figure 20c.

#### 3.2.2. Blade Deformation of the Proposed Combined Propeller-Blade Design

In Figure 20c, the final combined design twists approximately four times more than the reference case. The combined design also deflects less than the reference model. In addition, the final design eases off the twist at the tip as intended. A relative comparison of the reference case and the proposed combined design is given in Table 6. The absolute values in Table 6 are generated from the results presented in Figure 20b,c. Only a few radii are investigated for conciseness.

In Table 6, the blade design shows slightly better bend–twist properties for the maximum load case than for the minimum load case, indicating that the blade bends and twists more efficiently when overloaded. The effect is, however, not very pronounced. It is worth mentioning that the twist of the proposed design is 42.5 times larger than for the reference model at radius 0.5 because the reference shows almost no twist at this radius.

Ultimately, the blade twist should counteract the difference in the inflow angle between the two load cases, as stated in Section 1.3. In other words, the blade twist should match the difference in how the inflow changes on the propeller blade causing the load variation. An evaluation of how well the designs counter the inflow variation is given in Table 7.

Table 7 shows that the blade twist deformation at a radius of 0.7 counteracts slightly more than half the inflow variation at this radius while countering almost twice the inflow at a radius of 0.9. However, these numbers are based on simulations with a one-way FSI, which do not consider how the blade deformation would mitigate the loads and the blade deflection and twist nor how the material stresses might redistribute. Thus, the results can be regarded as indicative of the blade’s performance.

## 4. Discussion

This work aimed to explore existing and new design concepts for making adaptive composite marine propeller blades that maximize bend–twist coupling. Previous studies have found that propeller blades with bend–twist deformation characteristics could dissipate blade loads during operation. This paper explored four design concepts and proposed a propeller-blade design that showed deformation characteristics that should dissipate a load variation due to the propeller operating in a periodic-load case.

Insight into what deformation characteristics could be achieved with the different design concepts was gained by making exploratory blade models based on specific design concepts. For each design concept, three design choice models were made to observe what design characteristics one could expect from applying the different concepts. The design concepts were explored using a simplified flat square NACA0009 blade to limit the geometric factor of the blade and to generalize the findings so that they can be applied in other propeller-blade geometries later. With only a few design choices within each concept, this paper intends to show what can be expected when applying the various design concepts.

The novel aspects of this paper are that design concepts outside the published materials were considered for propeller-blade buildup. While several previous works have focused on the design concept of anisotropic surface laminates, this paper acknowledges that this design concept is only one of the possible measures that can be taken to design adaptive propeller blades.

The final NACA blade design that applied all the proposed design concepts showed that it was possible to achieve superior deformation characteristics in a blade using several design concepts rather than just one concept. The relative deformation characteristics of all NACA blade models are summarized in Table 4 to quantitively compare the effects of the different design concepts and choices.

After investigating the design concepts, the concepts were applied in a product development case for a commercial propeller blade with a typical geometry. The aim was to propose a material layup composition for the blade that caused deformation characteristics that would mitigate periodic-load variations. A final composite propeller-blade design for the commercial propeller-blade geometry was made that applied all the design concepts and achieved a significant twist. While the final design shows desirable deformation characteristics, it was not in any way optimised.

Previous research has identified the change in the water inflow angle on the propeller. An assumption was that the load variation could be mitigated if the blade twisted to somewhat counteract the inflow angle on the propeller blade, which it managed, as shown in Table 7.

In Table 8, the blade of Kumar and Wurm [5] is compared to the blade design proposed in this paper. The bend–twist characteristics of the proposed design exceed the bend–twist characteristics of the Kumar and Wurm blade. The blade in this study is half the diameter of the Kumar and Wurm blade, which affects the blade-twist effect on twist as the twisting manifests throughout the blade length when it bends, implying a much better bend–twist coupling.

This work focused on modelling structural design options for blades. Therefore, experimental verification of the blade’s structural response, computational fluid dynamics with two-way FSI to estimate the hydrodynamic performance properties and experimental verification of the hydrodynamical properties were out of this scope. Nevertheless, it is reasonable to assume that the final design and a physical prototype of the design would show similar deformation characteristics, as a previously proposed propeller-blade model using the same design method had been experimentally verified with regard to the structural elastic response [27,28,29,37]. Ultimately, the blade twist in the proposed design is more prominent with regard to the absolute values and bend–twist coefficient than in previous studies.

As no two-way FSI was performed on the design, the hydrodynamical properties are ambiguous, but the following can be rationalized: the Kumar and Wurm blade showed a thrust dissipation of 7% [5]. Because the proposed propeller design under minimum load twists slightly less than the Kumar and Wurm design, the dissipation of the blade thrust was expected to be slightly less than 7%; for example, 5%. However, the proposed design twists slightly more under maximum load, meaning that the thrust dissipation is slightly larger; for example, 10%. A 5% thrust dissipation of the minimum blade thrust changes the minimum thrust from 15.2 kN to 14.4 kN. On the other hand, a 10% dissipation of the maximum blade thrust changes the maximum thrust from 29.4 kN to 26.5 kN. Thus, these thrust dissipations make the load variation on the propeller drop from 14.2 kN to 12.1 kN, which means that it is reasonable to assume that approximately 15% of the load variation could be mitigated with the proposed propeller-blade design.

To fully describe and quantify the hydrodynamic properties of the proposed propeller-blade design, further studies must be performed in collaboration with hydrodynamic researchers to allow for two-way FSI simulations. In addition, future studies should preferably include experimental verifications, which would require one to consider manufacturability, experimental design and numerical work. These have been conducted and can be performed again [5,10].

While the mitigation of the load variation could be improved further by improving the propeller-blade design, it is believed that other measures apart from altering the buildup should also be considered. For example, the propeller-blade geometry used in this paper was designed for a rigid metal propeller blade. In other words, this paper has built a flexible propeller blade with rigid propeller-blade geometry, which is probably not ideal. In addition, because a flexible bend–twist blade deflects and twists from loading, there would be a geometry change with a twist between the unloaded configuration and cruising load for a flexible propeller blade that is not relevant for rigid propeller blades. Therefore, the shape of a composite propeller blade would be different from a metal propeller because it should consider both its geometry during operation and just after production.

Another observation was that the change in inflow angle was small at large radii and large at small radii. The Kumar and Wurm blade [5] and the blade design proposed in this paper show the opposite behaviour with a small twist at small radii and a large twist at larger radii. One plausible solution to this discrepancy in the desired twist and achieved twist could be to focus on in-duct propellers with bend–twist properties that allow for twist rotation around the connection to the blade hub. However, exploring and proposing such composite propeller-blade geometries is both advanced and out of the scope of this paper.

This paper examines the enormous possibilities when composing design layups and considers the propeller design’s strength. Two of the six topics that Young et al. recently stated should be considered when moving towards next-generation bend–twist propellers [3]. An obvious next step for the propeller-blade design would be a two-way FSI to properly evaluate the thrust dissipation of the proposed combined design. The design should also be simulated in several other common operational scenarios, not just the operational scenario in focus here, to determine if the design has sufficient strength to endure the expected use.

## 5. Conclusions

Passive bend–twist deformation of composite propeller blades can be designed through the design concepts of making hollow-shell structures, using anisotropic laminates, applying local reinforcement patches, inserting inner reinforcement structures and removing the tail part of the root connection.

The main design concepts were first investigated for the simple geometry of the NACA blade, making the findings generalized for other blade designs. The deformation characteristics of the blade were identified by plotting the deformations under a load of the various design choices. The design concepts caused various amounts of blade twists and different bend–twist characteristics. The feasibility of the most promising concepts was explored by ensuring sufficient strength with maximum strain failure criteria.

A final design that utilized all the design concepts was made for the simple geometry of the NACA blade and the propeller-blade geometry. Good bend–twist deformations were achieved for both propeller blades, exceeding some of the hydrodynamic requirements throughout the blade span. For the typical blade geometry, the twist at the blade tip was 1.4°, which was in the same range as the angle change in the fluid that caused periodic-load variation. From a hydrodynamic point of view, it would be optimal to have the most twisting close to the root of the blade. It was possible to move significant twisting closer to the root even though the bending was quite small at that position.

Since the design methods to obtain maximum bend–twist coupling shown here worked for two very different blade geometries, they are likely to be applicable to any blade geometry. The designer has now a toolbox to create the maximum or desired bend–twist coupling for a particular application. Real physical experiments should be made to confirm the calculations.

## Figures and Tables

**Figure 1 polymers-15-02749-f001:**
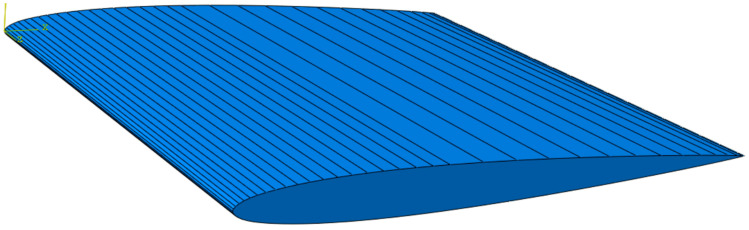
A simple blade profile made from a NACA0009 profile to test the design concepts on a general structure. The lines on the blade indicate the geometrical features; they do not represent the mesh.

**Figure 2 polymers-15-02749-f002:**
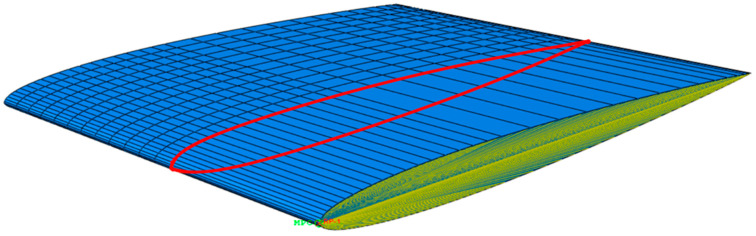
The NACA blade model is viewed from the leading-edge root perspective. A NACA0009 foil-shaped cross-section at 40% blade length is highlighted in red. These different cross-sections were tracked along the blade during blade loading to observe and quantify the deformation characteristics. The red text, RP-1, indicates the reference point that with the small green text, MPC, and the yellow lines makes up the BC.

**Figure 3 polymers-15-02749-f003:**
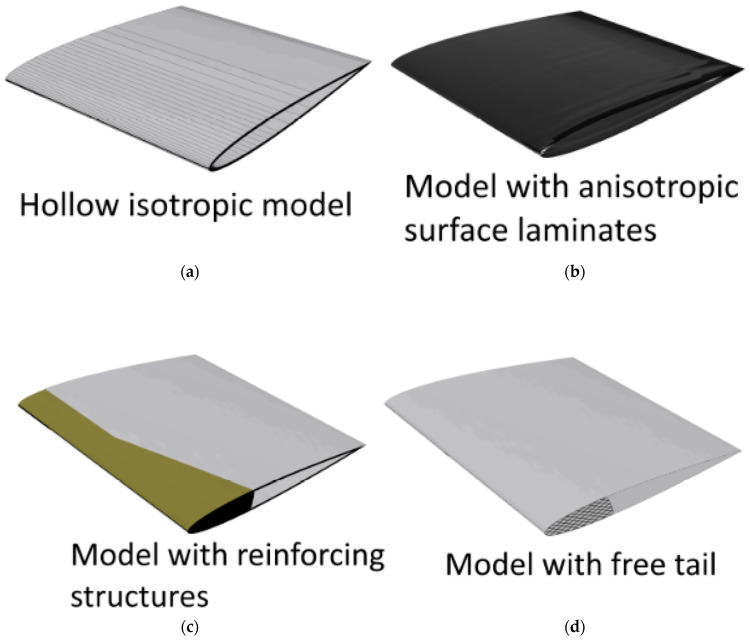
Model illustrations based on the investigated design concepts: (**a**) a hollow aluminium blade and (**b**) a hollow composite blade with anisotropic surface laminates. (**c**) is a blade made from a solid brass leading edge as an internal structure with a hollow blade tail and (**d**) illustrates a solid aluminium blade with no root fixture on the blade trailing edge freeing the blade’s tail movement.

**Figure 4 polymers-15-02749-f004:**
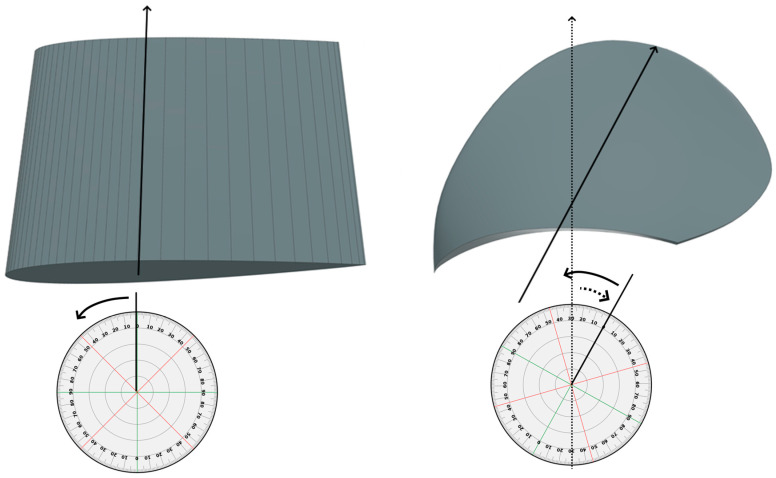
The NACA blade has a square frame and a symmetric foil profile throughout the length. The propeller blade has a more complex shape; for example, a skew, changing the angle of the rotation centre through the blade-tip line. The propeller rotation direction is indicated by the solid arrow, the change in blade angle is indicated by the dotted arrow.

**Figure 5 polymers-15-02749-f005:**
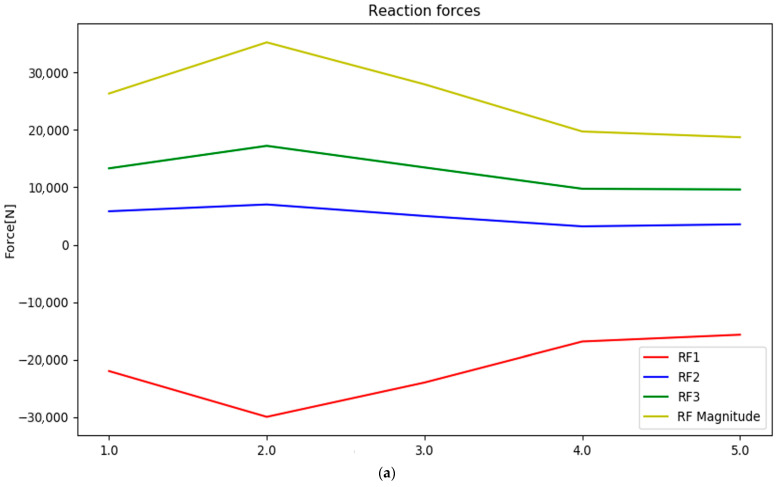
(**a**) shows the periodic variation that should be designed for. (**b**,**c**) Illustrate the pressure distributions in the maximum load case (**b**) and the minimum load case (**c**) under periodic loading. The pressure is highest in the red and orange areas and lowest (due to suction) in the blue and light green areas, as shown in the legends.

**Figure 6 polymers-15-02749-f006:**
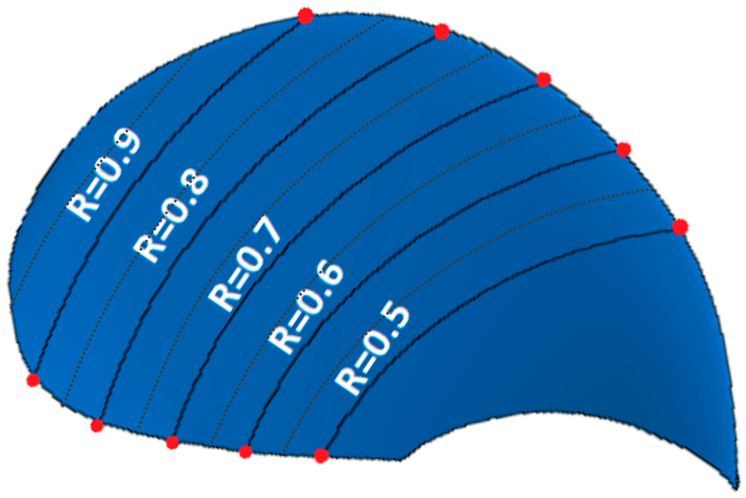
The circular lines were used for data collection to view the blade deformation characteristics. Figure from [27].

**Figure 7 polymers-15-02749-f007:**
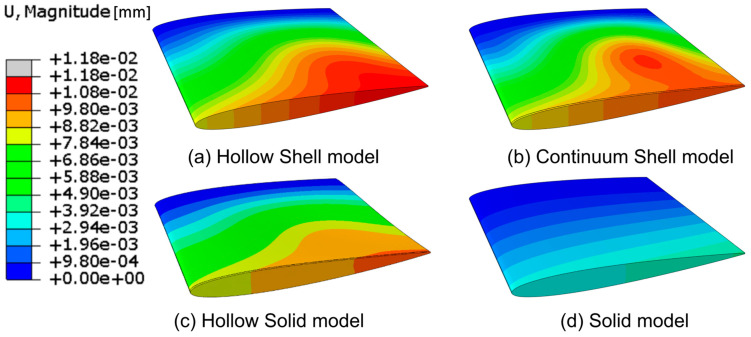
Blade displacement plots of the hollow and solid NACA blade models. The blades are viewed from the tip side, and the BC is on the far side. The graphs do not show that they are hollow but instead focus on how the models deform. The solid model (**d**), as expected, deforms to only a fraction of the other models. The shell model (**a**) is the most compliant of the hollow blade models (**a**–**c**).

**Figure 8 polymers-15-02749-f008:**
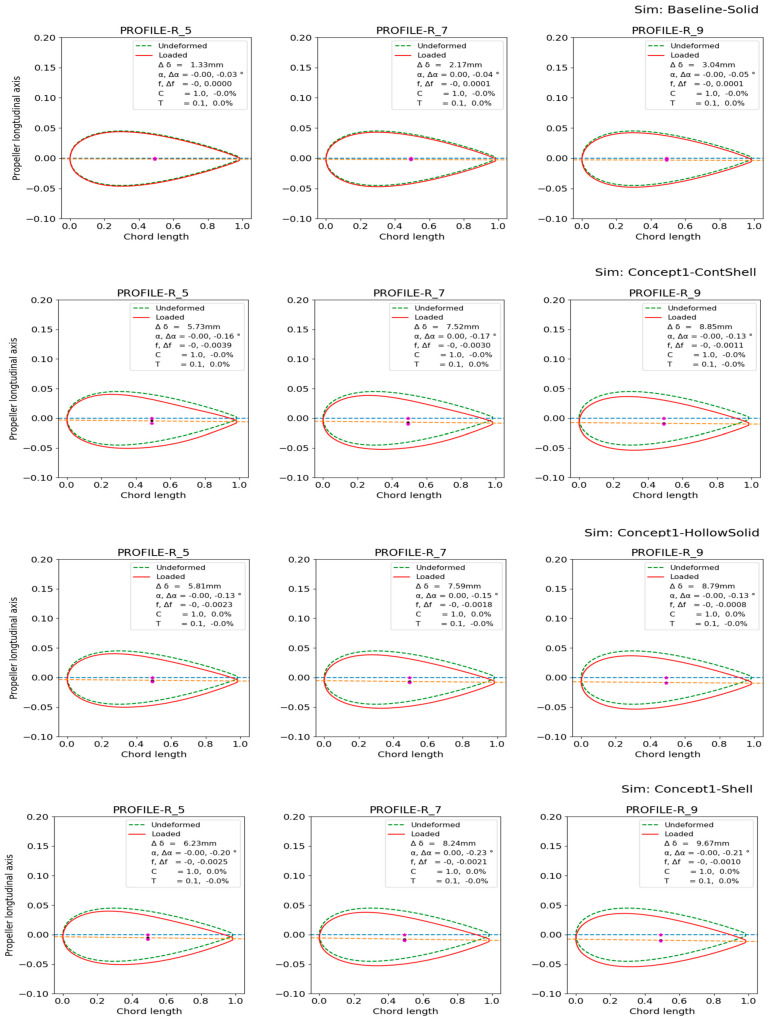
The foil cross-sections of the deformed and undeformed NACA beam baseline models. The cross-sections at 50%, 70% and 90% of the blade length are shown. The deflection (δ), twist (α), camber (f), chord length (C) and thickness (T) were tracked for each cross-section. The blue dotted lines are the undeformed chord lines while the orange lines are the chord line of the deformed foil.

**Figure 9 polymers-15-02749-f009:**
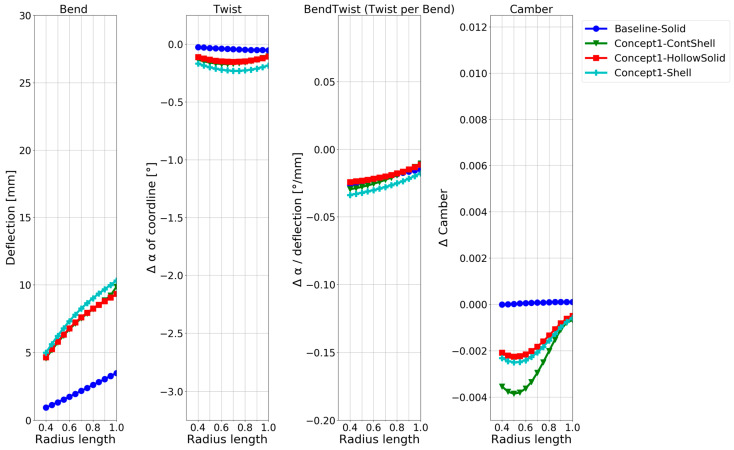
Hollow-shell models’ deformation characteristics compared to the baseline case.

**Figure 10 polymers-15-02749-f010:**
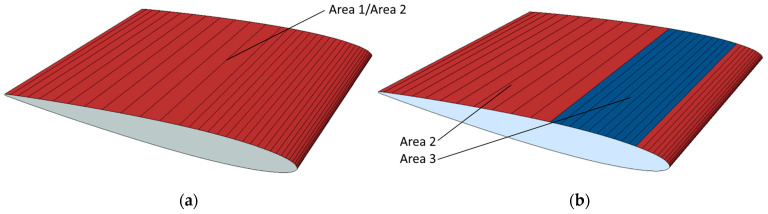
First and second models used only plies that covered the entire surface, indicated by illustration (**a**). (**b**) shows the third model with a local patch, a flange, on both sides of the location indicated in blue. The areas in the figure refer to Table 2.

**Figure 11 polymers-15-02749-f011:**
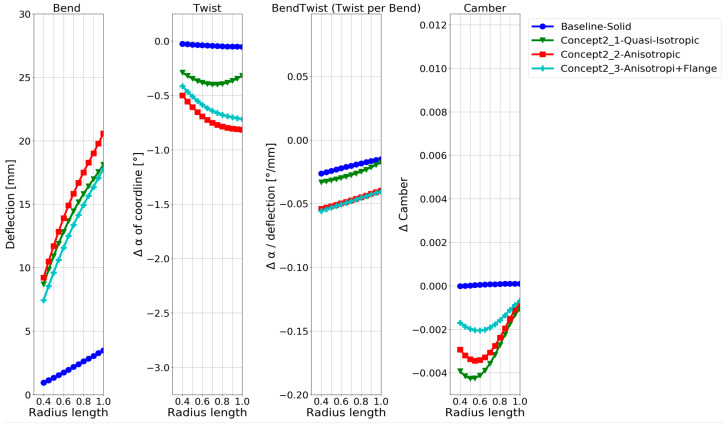
Comparison of the deformation characteristics of hollow anisotropic composite NACA blade models and the baseline case.

**Figure 12 polymers-15-02749-f012:**
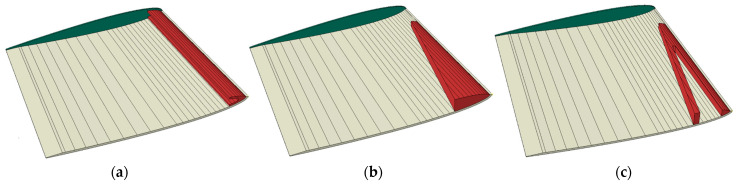
The geometry of the internal structures investigated in the NACA blade. The internal structures were made of titanium, and the hollow blade was made of aluminium. (**a**) was called LE-mast, (**b**) was LE-cornerstone and (**c**) was LE-framework.

**Figure 13 polymers-15-02749-f013:**
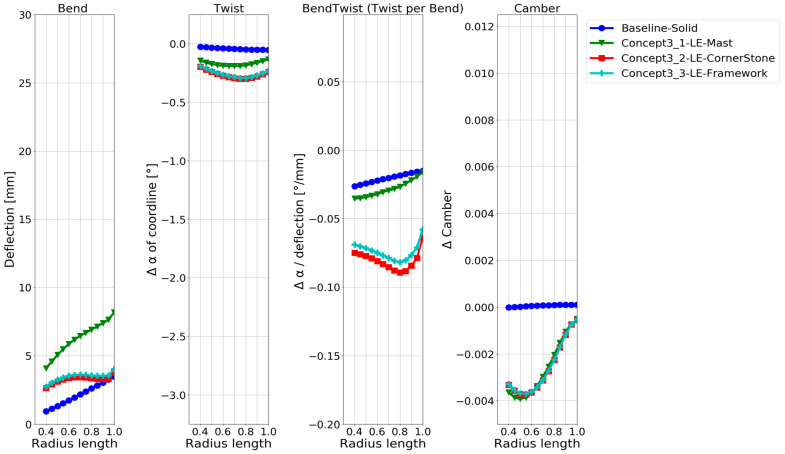
Comparison of the reinforcing-structure blade models and baseline model of a solid NACA blade.

**Figure 14 polymers-15-02749-f014:**
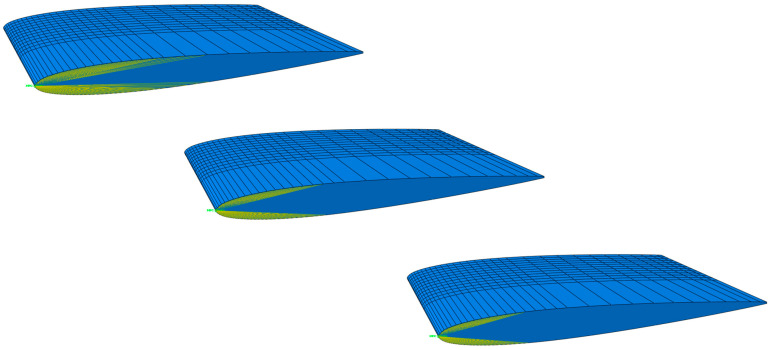
Graphic representation of how the BC was released from the tail of the NACA blade to achieve bend–twist deformation. The small green text indicates the MPC reference point.

**Figure 15 polymers-15-02749-f015:**
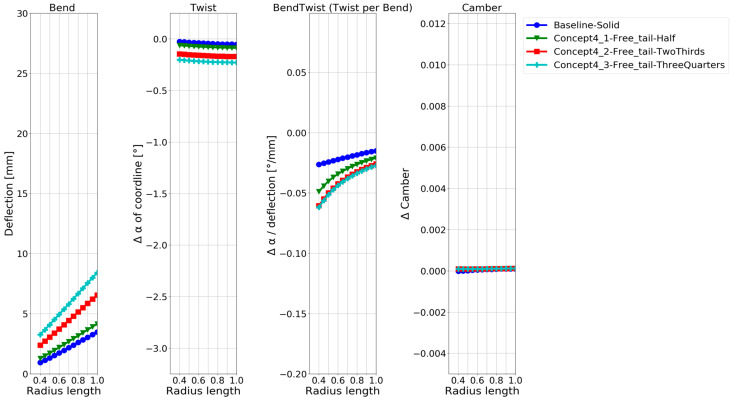
Comparison of the models that free up the tail to obtain bend–twist deformation. A solid isotropic model is plotted as the baseline.

**Figure 16 polymers-15-02749-f016:**
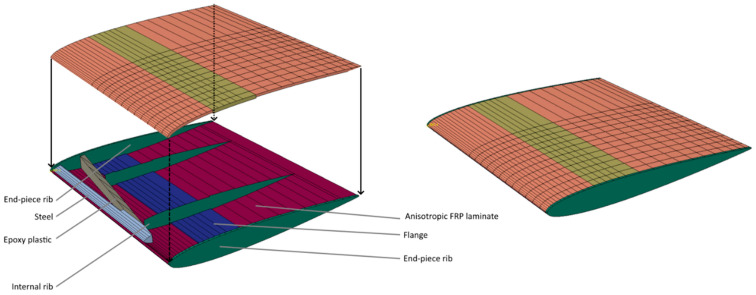
Composition of the “combined design concepts” for the proposed NACA blade design. The solid reinforcing structure was made of steel and epoxy and was used in combination with two internal FRP ribs. In addition, a hollow anisotropic FRP surface shell and FRP top and bottom end-piece caps are used. Finally, 66% of the tail was freed up. Some product development that explores assembly and fastening methods will be needed for this blade design to have a physical prototype.

**Figure 17 polymers-15-02749-f017:**
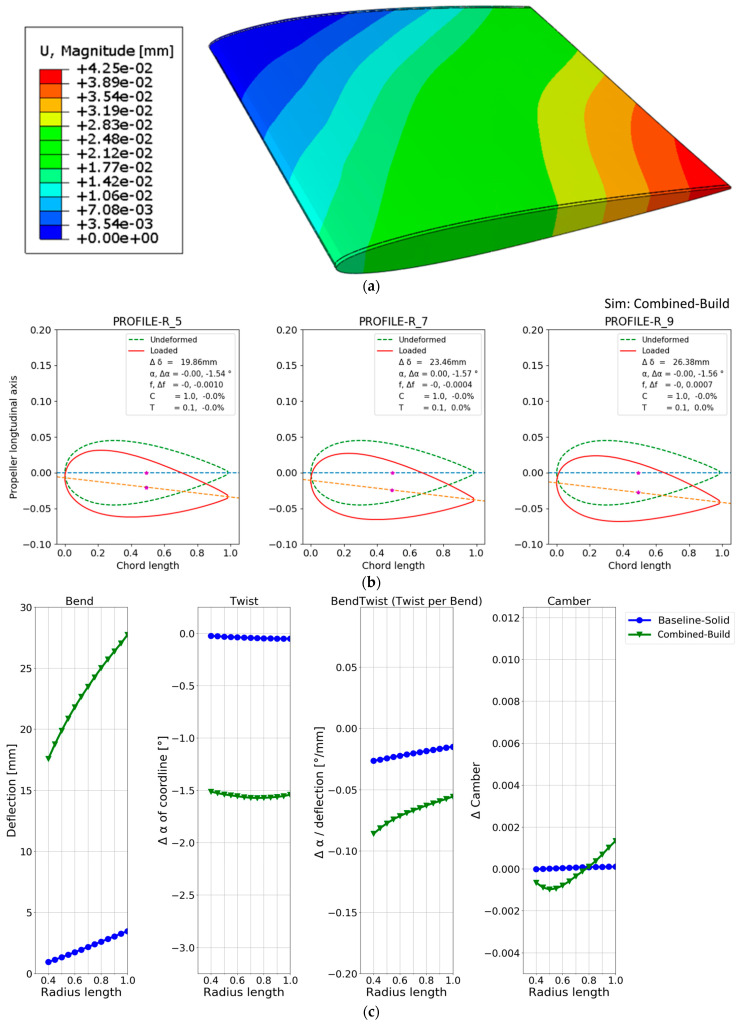
Global deformation contour of the combined NACA blade design in (**a**) and the deformation characteristics of the design at radii of 50%, 70% and 90% in (**b**). (**c**) compares the deformation characteristics between the combined NACA blade and the baseline case. The blue dotted lines are the undeformed chord lines while the orange lines are the chord line of the deformed foil.

**Figure 18 polymers-15-02749-f018:**
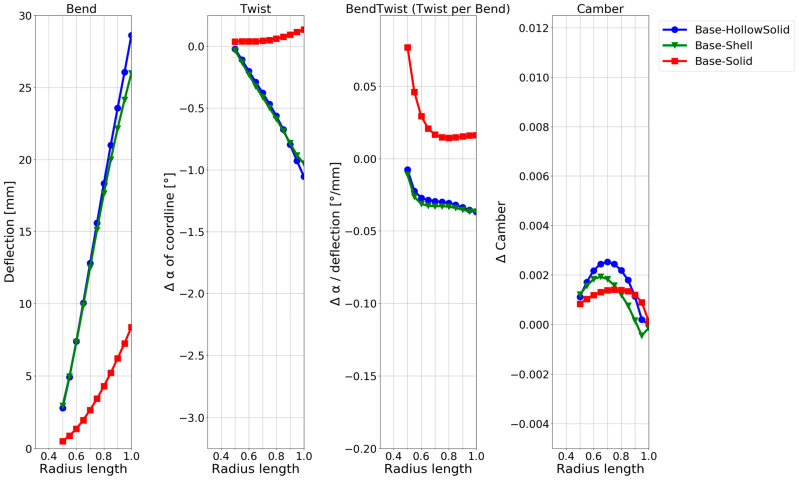
Comparison of the deformation characteristics of the solid isotropic model and the hollow isotropic models loaded with the maximum periodic load.

**Figure 19 polymers-15-02749-f019:**
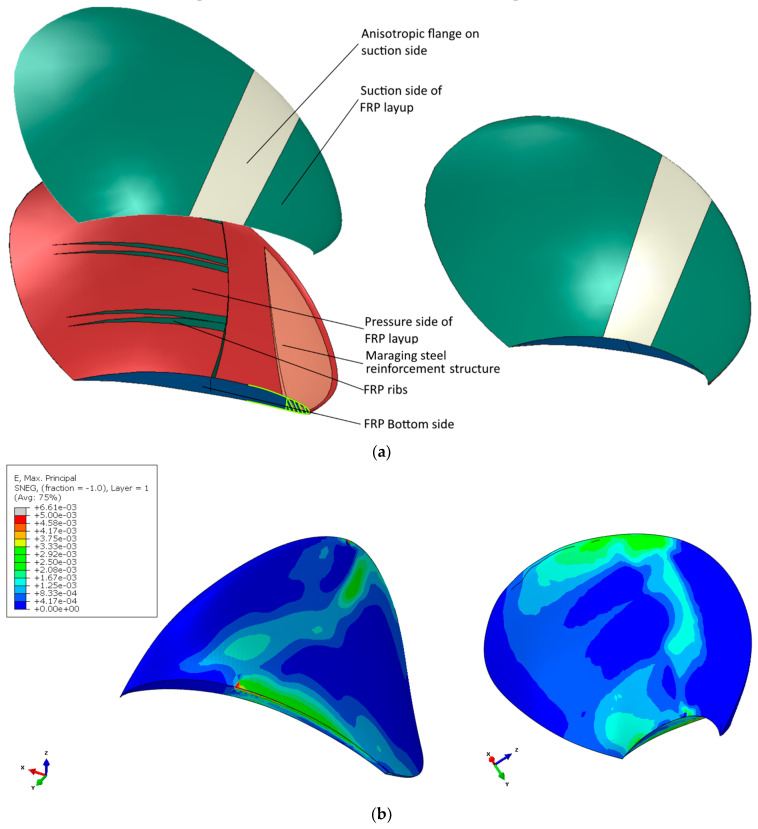
(**a**) shows the material composition in the combined propeller-blade design with the BC indicated in yellow. (**b**,**c**) shows the FEA estimation of the strain field in the blade designs’ composite surfaces and the blade’s internal substructures under maximum loading.

**Figure 20 polymers-15-02749-f020:**
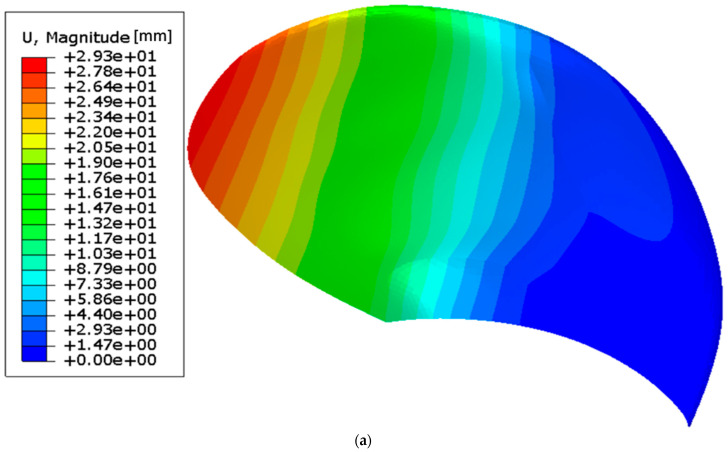
The global displacement contour of the typical propeller-blade design is shown in (**a**), and the deformation characteristics of the design at radii of 50%, 70% and 90% for both load cases are shown in (**b**). In (**c**), a comparison plot of the deformation characteristics of the final and baseline reference designs is shown.

**Table 1 polymers-15-02749-t001:** Material properties used to model the FRP materials.

FRP Ply Material	E1 [MPa]	E2 [MPa]	E3 [MPa]	ν12	ν13	ν23	G12 [MPa]	G13 [MPa]	G23 [MPa]	Thickness [mm]
CFRP Woven [0,90]	58,000	58,000	7500	0.05	0.3	0.3	3500	3300	3300	0.46
CFRP UD	117,000	7500	7500	0.34	0.34	0.5	3500	3500	3300	0.3
GFRP Woven [0,90]	26,000	26,000	8000	0.1	0.25	0.25	3800	2800	2800	0.4

**Table 2 polymers-15-02749-t002:** Layup of the anisotropic NACA models. The layup angles in dark red text are CFRP [0,90] plies, and the plies in black text are CFRP UD plies. The * indicates extra flange plies.

Model	Area	No. Plies	Layup	Maximum Strain
Quasi-isotropic	1	22	[0,45,0…45,0,45]	3.41%
Anisotropic	2	26	[0,70,30,20,30,20,70,30,20,30,20,30,0,30,0,30,20,30,20, 30,20,30,20,30,70,0]	4.90%
Extra flange patch	3	26 + 9*	[0,70,30,20,0*,30*,30,20,20*,70,30,30*,20,70,30,20,30,0, 70*,30,0,30,20,30,20,30*,30,20*,20,30,20,30*,0*,30,70,0]	4.76%

**Table 3 polymers-15-02749-t003:** Laminate stack description for the composite components in Figure 15. Only CFRP was used in this design. The layup angles in dark red text are CFRP [0,90] plies, and the plies in black text are CFRP UD plies. The * indicates extra flange plies.

Component	No. Plies	Layup
Anisotropic FRP laminate	25	[0,20,30,20,30,20,30,20,30,0,30,70,30,70,30,0,30,20,30,20,30,20,30,20,0]
Extra flange patch plies	25 + 12*	[0,20,70*,30*,30,20*,20,30,20,30,30*,20,30,0,30*,0*,30,70,30,70, 30,0*,30*,0,30,20,30*,30,20,30,20,20*,30, 30*,70*,20, 0]
Internal rib	22	[0,45,0,45,0,45,30,20,60,70,0,0,70,60,20,30,45,0,45,0,45,0]
End-piece rib	15	[0,45, 0,45, 0,45, 0,45, 0,45, 0,45, 0,45, 0,45]

**Table 4 polymers-15-02749-t004:** Relative deformation characteristics for several design cases compared with the standard solid metal blade. Relative values are used because both the NACA blade geometry and the load case are arbitrary. As the values are relative to the reference case, they are unitless.

NACA Models		Deflection			Twist			Twist per Deflection			Camber Change (Foil Shape)
	Radius	0.5	0.7	0.9	0.5	0.7	0.9	0.5	0.7	0.9	
Solid isotropic	Relative reference case	1	1	1	1	1	1	1	1	1	Very small
Hollow isotropic	Continuum shell	4.3	3.5	2.9	5.3	4.3	2.6	1.16	1.15	0.88	Small
	Hollow solid	4.4	3.5	2.9	4.3	3.8	2.6	0.96	1	0.88	Very small
	Shell	4.7	3.8	3.2	6.6	5.8	4.2	1.3	1.4	1.29	Very small
Anisotropic laminates	Quasi-Isotropic	8.2	6.6	5.6	11.6	10	7.4	1.3	1.4	1.29	Medium
	Anisotropic	8.8	7.3	6.3	20.3	18.8	16	2.16	2.3	2.5	Medium
	Anisotropic + local flange	7.2	6.2	5.4	17	16	14	2.21	2.4	2.5	Small
Reinforcing structures	LE-mast	3.8	3	2.4	5.6	4.8	3.2	1.4	1.45	1.3	Medium
	Cornerstone	2.3	1.6	1.09	8	7.3	5.6	3.2	4.3	4.9	Medium
	Framework	2.4	1.7	1.16	7.6	7.3	5.4	3	3.9	4.5	Medium
Solid isotropic with free tail	50%	1.3	1.2	1.2	2.3	2	1.8	1.6	1.5	1.35	Very small
	66%	2.3	2	1.9	5	4	3.4	2.1	1.9	1.7	Very small
	75%	3.1	2.7	2.5	7	5.5	4.6	2.17	1.9	1.8	Very small
Combined design	All design concepts combined	14.8	10.8	8.7	51.3	39.2	31.2	3.2	3.4	3.5	Medium

**Table 5 polymers-15-02749-t005:** Laminate stack description for the composite components in Figure 19. The layup angles in dark red text are CFRP [0,90] plies, the blue text is GFRP woven [0,90] plies, and the plies in black text are CFRP UD plies. The * indicates extra flange plies.

Area	No. Plies	Layup
Suction side	11	[45,30,20,30,45,30,45,30,20,30,45]
Extra flange area	15 + 4*	[45,20,10,20,10*,20*,45,20,45,20*,10*,20,10,20,45]
Pressure side	15	[−20, −30, −60, −30, −20, −30, −45, −30, −45, −30, −20, −30, −60, −30, −20]
FRP Bottom side	15	[0,45,20, −20,45,0,45,0,45,0,45, −20,20,45,0]

**Table 6 polymers-15-02749-t006:** Relative deformation characteristics for the proposed design compared to the standard hollow-solid metal blade. Because the values are relative to the reference case, they are unitless and represent the bend–twist efficiency.

Propeller Blade Models		Deflection			Twist			Twist per Deflection			Camber Change
	radius	0.5	0.7	0.9	0.5	0.7	0.9	0.5	0.7	0.9	
Hollow-solid isotropic blade, reference case	Maximum periodic load	1	1	1	1	1	1	1	1	1	Small
	Minimum periodic load	0.55	0.55	0.54	0.5	0.55	0.6	0.88	1	1.08	Small
Typical propeller blade	Maximum periodic load	1.96	0.91	0.88	42.5	4.92	4.3	17.9	5.3	4.85	Large
	Minimum periodic load	1.09	0.51	0.49	23	2.7	2.4	16.9	5.3	4.8	Large

**Table 7 polymers-15-02749-t007:** Comparison between achieved pitch change and change in apparent angle at radii of 0.5, 0.7 and 0.9.

	R = 0.5	R = 0.7	R = 0.9
Change in pitch	0.41°	0.85°	1.52°
Change in inflow angle	2.3°	1.3°	0.8°
% countered variation	17%	65%	190%

**Table 8 polymers-15-02749-t008:** Deformation characteristics of the proposed propeller design under chosen load cases and the Kumar and Wurm propeller blade under cruising load.

Propeller-Blade Models	Deflection [mm]			Twist [°]			Twist per Deflection [°/mm]		
radius	0.5	0.7	0.9	0.5	0.7	0.9	0.5	0.7	0.9
Kumar and Wurm design	15	31	47	0.5	1.3	2.3	0.03	0.04	0.05
Proposed design under minimum load	2.6	5.5	10	0.5	1.0	1.8	0.19	0.18	0.18
Proposed design under maximum load	4.7	10	18.6	0.9	1.8	3.3	0.19	0.18	0.18

## Data Availability

The data presented in this study are available within the article.

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
