# Peer review of "Designing Composite Adaptive Propeller Blades with Passive Bend–Twist Deformation for Periodic-Load Variations Using Multiple Design Concepts"

_polymers, 2023, doi:10.3390/polym15122749_

Round 1

Reviewer 1 Report

Title: Designing Adaptive Propeller Blades With Passive Bend-Twist Deformation For Periodic Load Variations Using Multiple Design Concepts

Reviewer comments: 

This manuscript has been the focus on Designing Adaptive Propeller Blades With Passive Bend-Twist Deformation For Periodic Load Variations Using Multiple Design Concepts. The study is original however I feel that the paper could be improved. Therefore, could you consider some points below for further improvement.

1.          Tittle:

Overall, the title effectively conveys the subject matter, scope, and key aspects of the study, generating interest and anticipation for the research findings. Good job!!

2.          Abstract:

Overall, the abstract effectively communicates the objectives, methodology, and key findings of the research. It generates interest by highlighting the improved bend-twist efficiency and the potential benefits of the design for mitigating adverse effects during operation. Great!!

3.          Introduction:

The introduction can become more coherent, clear, and engaging for the readers, providing a solid foundation for the rest of the manuscript. Please improve the below remarks. 

o    The introduction contains a lot of information but lacks a clear structure. Please improve the readability and organise introduction writing / paragraph wisely (introduction structure, i.e., Intro > Problem statement > Research gap > Objective & motivation)

o    Some sentences are lengthy and complex, which can make the content difficult to understand. Please improve the sentences into shorter, more concise to improve clarity and readability.

o    What are the relevance and importance of the concept mentioned such as one-way and two-way fluid-structure interaction modelling and optimization of ply orientation. Please improve for further explanation to help readers understand!!

o    There are typos and grammatical errors throughout the text. Make sure to proofread the introduction carefully to ensure correctness!!

4.          Material & Methodology (Determination of material characteristics):

The writing in this part is creative and packed with information. Although the author is conducting an excellent study, the framework could be improved by taking into account the comments below:

o  Design methodology:

·     Highlight the applicability of the chosen FEA methodology to the propeller blade geometry and its grounding in reality through experimental verification.

o  FEA simulation setup:

·     Provide details on the FEA simulation setup, including mesh sensitivity studies and element selection.

·     Explain the rationale behind using different element types to model the same designs and the significance of exploring their influence.

o  Design performance evaluation:

·     Outline the criteria and metrics used to evaluate the performance of the different design concepts.

·     Describe the analysis of deformation characteristics, intrinsic properties, and performance estimates for each design concept.

·     Explain how the simulations will be used to identify promising design concepts.

5.          Results & Discussion:

The results and discussion section well-presented and analysed the findings from evaluating the adaptive propeller blade designs. It will allow for a comprehensive comparison of the different design concepts and their potential implications for practical applications. However, there are a few areas that can be improved:

o    Clarity and organization: Consider restructuring the paragraph to present the findings and key points in a logical order, making it easier for readers to follow the flow of information.

o    Lack of specific results: While the conclusion mentions that "good bend-twist deformations were achieved," it does not provide specific details or quantitative data to support this claim. Including specific measurements or results regarding the achieved deformations and their comparison to hydrodynamic requirements would add more substance to the conclusion. Please improve!!

o    Incomplete analysis: The conclusion briefly mentions the hydrodynamic perspective and suggests that having more twisting close to the root is beneficial. However, it does not delve into a comprehensive analysis or discussion of the hydrodynamic implications of the achieved bend-twist characteristics. Providing a more detailed analysis and discussing the potential effects on propeller performance and efficiency would strengthen the conclusion.

o    Limited scope: The conclusion focuses solely on the NACA blade geometry and propeller blade geometry. It would be beneficial to acknowledge the limitations of the study and discuss the potential applicability or generalizability of the findings to other blade designs or scenarios. This would provide a broader context and enhance the relevance of the research.

6.          Conclusion:

Overall, the conclusion of the academic manuscript provides a summary of the findings and highlights the design concepts that were explored to achieve passive bend-twist deformation of propeller blades. Please provide more specific and detailed findings, discuss the hydrodynamic implications of the achieved deformations, acknowledge any limitations, and consider the broader applicability of the design concepts.

7.          Plagiarism:

Detected 7%: Good job!! 

research.

There are typos and grammatical errors throughout the text. Make sure to proofread the introduction carefully to ensure correctness!!

Author Response

Dear reviewer:

Thank you for your helpful comments. We tried to implement the comments and feel the changes have improved the paper, especially the clarity.

Please find below answers to your comments in blue in the attached document. The changes in the paper are indicated by “track changes”.

Best regards

The authors

Reviewer 2 Report

The manuscript "Designing adaptive propeller blades with passive bend-twist deformation for periodic load variations using multiple design concepts" shows huge amount of information how blades can be combined with different shapes, twisted etc.

The main issue of this work it contains too much information for one research paper. Therefore the reviewer suggest to restructure the script because for readers following such concept with 20 Figures and 8 Tables can be confusing.

1. The introduction should be short as with 5 pages alone should contain state of the art and the goal of this work. What is novel in this work?  For the material and method most parts reads as introduction. Suggested some of those figures can go to supplementary. Please use for material and method only such what belongs there, as there are some discussion given in this section. Additionally the author used mainly modelling of such design. What are its limitation, please add such in this section. 

2. The Result parts shows the different concept with readable language its still a lot of information given. Please try to make it more compact as the content overall very interesting but needs more focused on one design type. Beside the modelling and simulation, are those blades tested in real experiments to verify how the model fits?

3. the different concept combined in each as shown in the discussion as well conclusion are certainly novel while it would be interested regarding the literature which of such design are already used in real. A Table of comparison would fit there to overall show this concept in comparison to others as shown in Table 8. It would be beneficial adding this work design in it.

4. There several typos in the manuscript as Figure 20 is shown as Figure 2, Please check your manuscript on typos and misleading sentences

The English is fine just minor spell checking as well some typos need to be corrected

Author Response

(The authors gave the same response as above.)

Reviewer 3 Report

This paper presented a detailed design of propeller blades under cyclic loads using multiple design methods. The authors first provided a comprehensive review of the state of the art. Then, the materials and design methods were presented with adequate details. The authors reported a significant amount of work in the results section. The conclusions supported the discovery of this study. The reviewer has a few minor comments and suggests a revision before publishing this paper. 

1. The paper includes a significant amount of results and is longer than most research articles. Please consider summarizing the introduction and moving some results to the supporting materials instead of keeping everything in the main content of this paper. 

2. Since this journal is a journal focusing on polymers, how different polymer composites impact the overall performance of the propellers should be emphasized. Additionally, any experimental results can strengthen the paper and improve the quality of this paper. 

The quality of English writing in this paper is good. The reviewer can easily follow the content of this paper. Some minor revisions and proofreading are suggested before publishing this paper. 

Author Response

(The authors gave the same response as above.)
